# Effectiveness of health promoting schools: A comparative health profile assessment of higher as compared to low accredited schools in Chandigarh, Union Territory of North India

Jarnail Singh Thakur[1]*, Meenakshi Sharma[1], Sukriti Singh[2], Bhavneet Bharti[3], Rupinder Kaur[1], Meenakshi Sharma[4]

**1** Department of Community Medicine and School of Public Health, Post Graduate Institute of Medical Education and Research, Chandigarh, India, **2** Centre for Sustainable Development, Health & Wellness, Punjab, India, **3** Advanced Paediatric Center, Postgraduate Institute of Medical Education and Research, Chandigarh, India, **4** Indian Council of Medical Research, New Delhi, India

* jsthakur64@gmail.com

## Abstract

**Data Availability Statement:** All relevant data are within the paper and its Supporting Information files.

### Objectives

To assess and classify all private and government schools located in a northern city of India for accreditation as health promoting schools and comparative health profile assessment of selected higher accredited schools with lower accredited and non-accredited schools

### Design

Quasi experimental study with pre and post assessment with comparison of higher with lower accredited schools.

### Settings

The current study was conducted in 206 schools of Chandigarh City of Northern India. Comparative health profile assessment was undertaken in 8 schools with 754 children from higher accredited (platinum, gold, silver) and 8 schools with 700 children from lower accredited (bronze) and non-accredited (below bronze) schools.

### Interventions

Multicomponent and multilevel intervention was undertaken with self-quality improvement by schools with help of a manual of accreditation of school as health promoting schools. Key intervention included capacity building, technical visits, supportive supervision, sensitization of policymakers and key stakeholders, implementation of policy initiatives, use of social media, technical support and monitoring of activities.

**Funding:** The author(s) received no specific funding for this work. The project was funded by ICMR,New Delhi but no funds for publishing the same was provided.

**Competing interests:** The authors have declared that no competing interests exist.

## Outcomes

Accreditation levels (bronze, silver, gold and platinum levels) as health promoting schools after pre and post intervention.

## Results

Out of 206 schools, 203 participated in the baseline assessment and 204 in the endline assessment. The response rate was 99%. Two schools which refused participation were excluded and not assessed. Schools (N = 17) which participated in the 2011–2013 study were excluded from analysis. There was a statistically difference (p = 0.01) in the improvement of accreditation level of the baseline and endline assessment after intervention (p<0.05). Overall, the proportion of schools at the gold level increased from 1(0.5%) in 2016 to 71(38%). Silver level from 9(5%) to 57 (31%) of schools after intervention. The response rate in health profile assessment in higher(8) and lower(8) accredited schools was 95.9% and 92.7% respectively. The health profile of children higher accreditation level schools (N = 754) were found better in hygiene practices protective factors (peer support at school, parental or guardian supervision), handling stress and less prone to injury as compared to lower accreditation level schools (N = 700),(p<0.05).

## Conclusions

The health promoting school programme was found to be feasible and effective and lead to significant improvement in accreditation level as compared to baseline assessment after continuous self-quality improvement by schools(p<0.05). The health profile of children studying in higher accredited schools was better as compared to lower accredited schools.

## Introduction

The health of children and adolescents is of supreme importance to the growth and development of any country. Along with the family, the school is one of the main settings in which individual and social development of children occur [1]. The interaction between school teachers and students provides a unique opportunity for health promotion that can be sustained and reinforced over time [2]. Hence, school is an appropriate setting to improve youth health.

The "Health Promoting School" (HPS) is a holistic approach to integrate health promotion within the community. HPS approach was inspired by the Ottawa Charter [3]. The concept of 'Health Promoting Schools' was adopted by the World Health Organization (WHO) in 1995, as part of a settings-based approach to health improvement. In Europe, North America, and the Western Pacific region, there have been significant developments in the promotion of children's health involving schools [4]. The Schools for Health in Europe network (SHE network) was introduced by the WHO for the European region with the Council of Europe and the European Commission [5].

Accreditation is a public recognition of the achievement of required standards by an organization [6]. It certifies the schools for their efforts in implementing health promotion initiatives following an assessment, further helping them in redeveloping and implementing effective HPS strategies. In India, a pilot study for accreditation of schools as 'Health promoting schools' was undertaken in Chandigarh in which a checklist and scoring indicators were

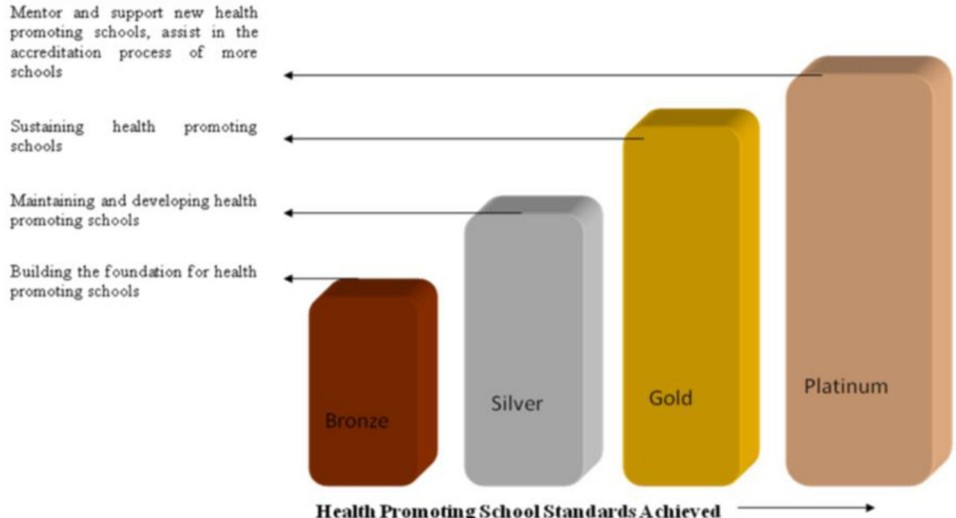

Mentor and support new health promoting schools, assist in the accreditation process of more schools

Sustaining health promoting schools

Maintaining and developing health promoting schools

Building the foundation for health promoting schools

**Fig 1. Health promoting schools accreditation standards.**

developed in 17 schools and was found to be feasible with active participation of WHO, Health Department of Chandigarh, Ministry of Health & Family Welfare, Govt of India, ICMR and Quality Council of India in 2010–11 [2]. Four categories of accreditation were developed namely, platinum, gold, silver, and bronze in the pilot study (Fig 1) [2].

The current study was conducted to assess and classify all private and government schools located in a northern city of India for accreditation as health promoting schools and comparative health profile assessment of selected higher accredited (platinum, gold, silver) schools with lower (bronze) and non-accredited (below bronze) schools.

## Materials and methods

### Settings and study design

A quasi-experimental study with pre-test and post-test design was conducted in 206 schools (public and private) in Chandigarh Union Territory of Northern India from September 2016 till May 2019 to evaluate the accreditation of schools as health promoting schools focusing on multicomponent and multilevel interventions before and after the intervention. All schools located in Chandigarh city were invited for participation by promoting voluntary participation and active involvement which was critical for the implementation of the interventions.

### Intervention

Interventions included self-quality improvements with the help of manual for accreditation of schools as HPS, sensitization of policymakers and key stakeholders, capacity building of the schools, technical visits, supportive supervision to schools, implementation of policy initiatives, use of social media, technical support and monitoring of activities.

**Manual.** Manual developed in an earlier pilot study (2011–2013) was reviewed and accreditation checklist consisted of eight domains: (Healthy School Environment, Mechanism for promoting health in schools, School health services, School nutrition services, Physical Education, School counseling, psychological and social services, Community partnership, Implementation of Shala Siddhi and mentoring schools in becoming HPS). The cut-off level of

scores for bronze, silver, gold, and platinum levels were used as 100–120, 121–150, 151–200 and >200 points, respectively, as per the standards developed in the previous study (Fig 1) [2].

**Key school-based interventions.**   Interventions for a period of 1 year (2017–2018) included zone wise division of all schools of the Chandigarh into three zones consisting of approximately 70 schools each: East, South and Central for providing technical support. After distribution of feedback reports on baseline assessment by zone coordinators to the schools; social media groups of respective zones were created.

Orientation and reorientation workshops for principals and teachers, convergence with other National health programs like Rashtriya Bal Swasthya Karyakram(RBSK)/Rashtriya Kishor Swasthya Karyakram(RKSK), school health program, mid-day meal and implementation of policy initiatives was done. Investigators coordinated with schools for mentoring of other schools; higher accredited schools on the basis of baseline assessment were encouraged to help lower accredited schools for which means of communication (exchange of contact details) was established between them. Investigators and key stakeholders also visited 10% of the schools for monitoring and supervision.

Schools those were categorized as gold and silver were classified as higher accredited schools and those classified as bronze and below bronze were classified as lower accredited schools. The above-mentioned categorization was decided in an expert group meeting considering the baseline results and comparison of basic characteristics (student teacher ratio, toilet ratios, water consumption person per day). For comparison between higher accredited and lower accredited schools, a sample size of 1400 was calculated by considering difference in prevalence of anemia among rural and urban school going children (14.16% vs 25.4%) with power of 80% and a design effect of 1.5 [5]. A multistage random sampling was performed. The first stage consisted of random selection of 8 schools from the higher (4 government and 4 private) and lower (4 government and 4 private) accredited the schools. Letters of invitation were sent to each selected school. In the second stage, we randomly selected one section of classes (7 and 8) from each selected school with an inclusion criterion of a student studying in the school for the last two years. In Chandigarh, one section of a class is comprised of 40 to 50 students and there are 4–5 sections of a particular class.

The health profile assessment was undertaken by GSHS questionnaire based on the adapted Central Board of Secondary Education (CBSE) Indian version of questionnaire [7] of 2007 with a focus on modules on diet, physical activity, tobacco use, alcohol use, drug use, mental health, hygiene, protective factors, violence and unintentional injury and sexual behavior modules. Only those students were covered who were available in the school on the scheduled day after obtaining written parental consent and the child's assent. The students were required to complete the questionnaires independently in 90 minutes (2 class periods).

Endline assessment was undertaken from October 2018–March 2019 onwards. Ethical permission for conduction of study was taken from the institutional ethics committee (PGI/IEC/ 2014/2217). The ranking of schools as HPS was released in a public function chaired by Education Secretary of UT Chandigarh making it the first state in India to do so in India.

## Data analysis

The data collected was entered in excel spreadsheet and analyzed by using IBM SPSS Statistics for windows, version 20.0, NY, and Epi info version 7.1.5, Atlanta, GA. The effectiveness of accreditation system of HPS was analysed by comparing the accreditation level of school at baseline and endline after intervention and comparing health profile of children studying in higher as compared to lower accredited schools. Initially, the data was screened for missing and outliers. Descriptive statistics was used to describe the study demographics using

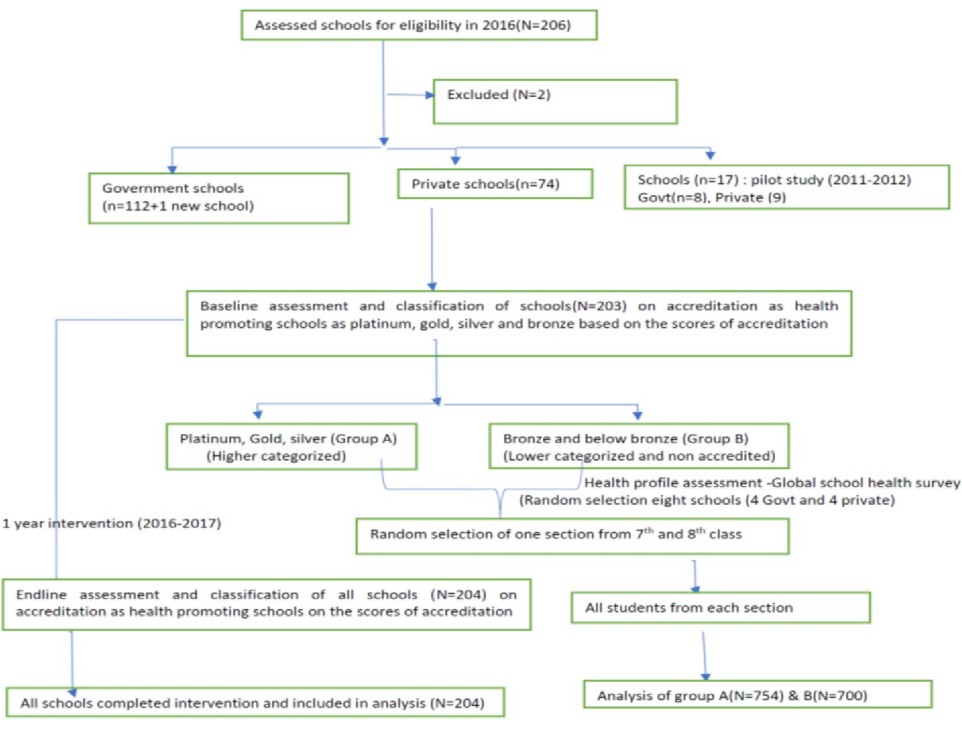

**Fig 2. Participating schools flow diagram.**

frequencies (n), percentages (%), means, and standard deviation (SD). Chi-square was used to examine differences among variables. All statistical tests were two-sided and statistical significance was defined at the 5% alpha level.

## Results

All schools in Chandigarh city (N = 206) were invited for participation in the study. Out of 206 schools, 203 participated in the baseline assessment and 204 in the endline assessment. Two schools which refused participation were excluded and not assessed as shown in the Fig 2. The response rate was 99%. Schools (N = 17) which participated in the 2011–2013 study were excluded from analysis [2]. Baseline characteristics of government schools (N = 113) and private schools (N = 74) were compared, and no major difference was found. Distribution of schools showed that there were more high schools (49%) as compared to senior secondary schools (32%) among government category (Table 1). Overall, there were more senior secondary schools as compared to other education levels in the private category. The median of toilet ratio for boys was 28.9(Inter Quartile Range (IQR) = 20–73) in the Government and 23 (16–39) in the private schools in 2018 (Table 1).

Table 2 depicted that one hundred fifty-two (82%) schools were below bronze level during the baseline assessment but after intervention the schools in the same level reduced to 25(13%) i.e., 16(14%) government and 9(12%) private schools respectively which was found to be significant(p = 0.01). The proportion of schools acquiring silver accreditation increased from 9 (5%) to 57 (31%) of schools (p = 0.01). Overall, the proportion of schools at the gold level increased from 1(0.5%) in 2016 to 71(38%) in 2018 which was found to be significant (p<0.05).

**Table 1. Distribution and key characteristics of Government and Private Schools in Chandigarh (2016–2018).**

| Category | Government (N = 113)* | Private (N = 74) |
|---|---|---|
| | n (%) | n(%) |
| **Education levels** | | |
| Senior Secondary | 37(32.7) | 43(58.1) |
| High | 55(49.1) | 12(16.2) |
| Middle | 13(11.6) | 9(12.1) |
| Primary | 8(7.1) | 10(13.5) |
| **Characteristics Median(IQR)** | | |
| Girls | 530(305–820) | 424(192–707) |
| Boys | 588(336–880) | 559(226–920) |
| Water consumption person per day | 17(11–23.9) | 16(10–32.1) |
| Toilet ratio-Boys | 28.9(20–7) | 23(16–39) |
| Toilet ratio-Girls | 41.5(26.6–56) | 30(18–42) |
| Student tap ratio | 52(38.6–85.7) | 50(36.9–79.7) |

*One new Govt school was functional in 2018. Figures in parenthesis are percentages.

The proportion of schools with school safety, security and presence of evacuation plan increased from 54% in 2016 to 92.5% in 2018 (Table 3). The presence of a School Health Committee (SHC) had significantly improved from 72% of schools in 2016, compared with 95.7% of schools in 2018(p = 0.02). Training for the HPS program in schools increased from 61% in 2016 to 89% in 2018(p = 0.01). The monitoring of canteens in schools for ensuring quality improved from 68% in 2016 to 99% in 2018 (P = 0.02). All schools included in the study were found to have appropriate playground facilities except 5 Government schools and 2 private schools. The designated hours each week assigned for physical activity (PT period) of 45 minutes per day for all days (minimum 5 days) increased to 96% compared to 90% in 2016 (Table 3).

Health profile of students from 8 higher accreditation schools (gold, silver) were compared in the lower (bronze) and below bronze accreditation schools with 754 children from higher and 700 from lower accreditation levels were selected for further health assessment. The response rate in higher and lower accredited was 95.9% and 92.7% respectively. There was no

**Table 2. Comparison of accreditation levels of Government and Private Schools in Chandigarh after one year of intervention (2016–2018).**

| Category / Level | Pre Intervention | | | Post Intervention | | | % age Change | P value |
|---|---|---|---|---|---|---|---|---|
| | Government schools (N = 112) n (%) | Private schools (N = 74) n (%) | Total (N = 186) n (%) | Government schools (N = 113) n (%) | Private schools (N = 74) n (%) | Total (N = 187) n (%) | | |
| Platinum | 0 | 0 | 0 | 0 | 2(2.74) | 2(1.1) | +1.1 | - |
| Gold | 0 | 1(1.3) | 1(0.5) | 39(34.5) | 32(43.8) | 71(38.1) | +37.6 | 0.01* |
| Silver | 5(4.4) | 4(5.4) | 9(4.8) | 39(34.5) | 18(24.6) | 57(30.6) | +25.8 | 0.01 |
| Bronze | 15(13.3) | 9(12.2) | 24(12.9) | 19(16.8) | 13(17.8) | 32(17.2) | +4.3 | 0.32 |
| Below bronze | 92(82.1) | 60(81.1) | 152(81.7) | 16(14.1) | 9(12.1) | 25(13.3) | -68.4 | 0.01 |

*pvalue, $X^2$.

**Table 3. Comparison of government and private Schools as per different domains of accreditation as Health Promoting Schools in Chandigarh after intervention (score >3***).**

| Domains | 2016 | | | 2018 | | | P value |
|---|---|---|---|---|---|---|---|
| | Schools (N = 186) | | | Schools (187) | | | |
| **I. Healthy School Environment** | Govt (N = 112) n(%) | Private (N = 74) n(%) | Total n(%) | Govt (N = 113) n(%) | Private (N = 74) n(%) | Total (N = 187) | |
| 1  Access to clean drinking water, clean toilets and adequate lighting** | 102 (91.0) | 67 (90.5) | 169 (94.9) | 107 (95.5) | 72 (98.6) | 179 (95.7) | 0.91 |
| 2  Sufficient dustbins for refuse disposal | 109 (97.3) | 72 (97.2) | 181 (97.9) | 112(99.1) | 73 (99.9) | 185 (99.5) | 0.95 |
| 3  School safety, security and presence of evacuation plan for which everyone is trained | 57 (50.8) | 42 (56.7) | 99 (53.5) | 106 (93.8) | 66 (90.4) | 172 (92.5) | 0.51 |
| **II. Mechanism for promoting health and awareness about Health Promotion in schools** | | | | | | | |
| 1  Presence of School Health Committee / related mechanism | 98 (87.5) | 36 (48.6) | 134 (72.1) | 109 (96.4) | 69 (94.5) | 178 (95.7) | 0.02 |
| 2  Knowledge and practices of personal hygiene | 105 (93.7) | 69 (93.2) | 174 (94.1) | 112 (99.1) | 72 (98.6) | 184 (98.9) | 0.91 |
| 3  Presence of notice board/walls | 94 (83.9) | 46 (62.1) | 140 (75.7) | 108 (95.6) | 72 (98.6) | 180 (96.8) | 0.18 |
| 4  Presence of posters and / or other means of publicizing and popularizing concept of Health Promotion in school and local community | 101 (90.1) | 53 (71.6) | 154 (83.2) | 112 (99.1) | 70 (95.9) | 182(97.9) | 0.44 |
| 5  Student awareness and understanding of Health promotion concept, objectives and strategies | 98 (87.5) | 64 (86.4) | 162 (87.6) | 112 (99.1) | 71 (97.3) | 181 (97.9) | 0.89 |
| 6  Training for Health Promotion Programmes | 86 (76.7) | 26 (35.1) | 112 (60.9) | 100 (88.5) | 65 (89.04) | 165(88.7) | 0.01 |
| 7  Presence of a coordinator/health incharge for the Health Promotion | 103 (91.9) | 48 (64.8) | 151 (81.2) | 111 (98.2) | 67 (91.8) | 178 (95.7) | 0.26 |
| 8  Curriculum which emphasizes on health related subjects and being taught | 105 (93.7) | 66 (89.1) | 171 (91.9) | 113 (100) | 71 (97.3) | 184 (98.9) | 0.99 |
| 9  Presence of sources and / or lectures on priority health subjects for students and staff | 102 (91.0) | 57 (77.0) | 159 (85.9) | 108(95.6) | 70(95.9) | 178(95.7) | 0.51 |
| **III. School Health Services** | | | | | | | |
| 1  Presence of health cards | 101(90.1) | 45 (60.8) | 146 (78.5) | 111 (98.2) | 67 (91.8) | 178 (95.7) | 0.19 |
| 2  Presence of first-aid kit | 107 (95.5) | 65 (87.8) | 172 (92.5) | 109(96.5) | 72(98.6) | 181(97.3) | 0.70 |
| 3  Training of students and staff on first aid | 63 (56.2) | 42 (56.8) | 105 (56.8) | 81(71.7) | 64(87.7) | 145(77.9) | 0.51 |
| **IV. School nutrition services** | | | | | | | |
| 1  Nutrition education in school | 98 (87.5) | 71(95.9) | 169 (91.4) | 112(99.1) | 72(98.6) | 184(98.9) | 0.58 |
| 2  Monitoring canteens/meals in the schools | 103 (91.9) | 25 (33.7) | 128 (68.8) | 110(97.3) | 50(98.0) | 160(99.4) | 0.02 |
| 3  Option of healthy food and drinks | 57 (50.8) | 27 (36.4) | 84(45.2) | 71(62.8) | 45(61.6) | 116(95.9) | 0.33 |
| **V. Physical education** | | | | | | | |
| 1  A minimum number of hours of physical activity per week to all students in or outside the school curriculum | 100(89.2) | 67 (90.5) | 167 (89.8) | 106(93.8) | 72(98.6) | 178(95.7) | 0.95 |
| **VI. School counseling, psychological and social services** | | | | | | | |
| 1  Presence of social programmes and counseling services | 81(72.3) | 40 (54.0) | 121 (65.05) | 95 (84.0) | 60(82.2) | 155(83.3) | 0.33 |

(*Continued*)

**Table 3.** (Continued)

| Domains | 2016 Schools (N = 186) | | | 2018 Schools (187) | | | P value |
|---|---|---|---|---|---|---|---|
| 2  Adolescent Education Programme services including life skill education | 76 (67.8) | 42 (56.7) | 116 (63.8) | 106 (93.8) | 67(91.8) | 173(93.0) | 0.58 |
| **VII. Community Partnership** | | | | | | | |
| 1  Community partners in decision-making and planning in the health promoting activities of the school | 94(83.9) | 49 (66.2) | 143 (76.9) | 113 (100) | 67(91.8) | 180(96.8) | 0.58 |
| **VIII. Extent of implementation of School standards and evaluation framework (Shaala Siddhi) and involvement in establishing more Health Promoting Schools and their accreditation** | | | | | | | |
| 1  Extent of implementation of SHAALA SIDDHI*** as per checklist | 91(81.2) | 6(8.1) | 97(52.4) | 101(89.4) | 8(10.9) | 109(90.2) | 0.74 |
| 2  Promoting and Assisting in the accreditation of other schools (Applicable in Midline or End line assessment). | 0 | 0 | 0 | 15(13.39) | 9(12.3) | 24(12.9) | |

** composite scoring of the parameters done as per HPS manual.

*** considered based on the previous published paper [2].

**** National Programme on School Standards and Evaluation by National Institute of Educational Planning and Administration (NIEPA), under the aegis of Union Ministry of Human Resource Development.

significant difference between demographic characteristics and as per gender among higher and lower accredited schools (p = 0.14). Among all the participants, majority of the students were in the age group 11–13 years i.e., 83% and 81% among government and private schools respectively.

Hygiene practices were better among students of higher accredited schools as compared to students of lower accredited schools (Table 4). More than 87% and 92% students from higher accredited schools reported that they washed their hands before eating and after toileting respectively as compared to lower accredited schools(p = 0.04).

Protective factors including peer support at school, parental or guardian supervision were found to significantly better among students of higher accredited schools (p = 0.01, p = 0.04). In relation to parents' or guardians' monitoring, 62% students of higher accredited schools reported approved parental control for observation on their free-time activities (Table 4).

Table 5 showed that more than 63% students from higher accredited schools reported that they were taught to handle stress in healthy ways as compared to 55% children in lower accreditation level during school year (p = 0.01).

The majority (66%) from lower accredited schools claimed they had serious injury happened to them in the past 12 months as compared to 62% higher accreditation level (p = 0.07) (Table 5).

About 20% students from higher accredited schools reported that parents or guardians drink alcohol as compared to 16% students from lower accredited schools(p = 0.05) (Table 5).

Students from lower accredited schools were more informed regarding HIV (36%) than those from higher accredited schools (46%). Similarly, it was found that students from lower accredited schools (30%) were more aware about HIV as compared to students from higher accredited schools (23%) (p = 0.01) (Table 5).

## Discussion

Th current study is a pre-post multicomponent and multilevel health promotion school accreditation intervention study of both government and private secondary schools in northern India. The aim was to compare the higher accredited schools versus lower accredited

**Table 4. Comparison of diet, physical activity and protective factors among school children in higher and lower accreditation schools in Chandigarh.**

| Results for students aged 11–16 years | Higher Accredited Schools | | | Lower Accredited School | | | p-value |
|---|---|---|---|---|---|---|---|
| | Total N(754) n% | Boys N(399) n% | Girls N(355) n% | Total N(700) n% | Boys N(391) n% | Girls N(309) n% | |
| **Dietary Behaviors and Overweight** | | | | | | | |
| Students who went hungry most of the time or always during the past 30 days | 45 (5.9%) | 23 (5.7%) | 22 (6.1%) | 55 (7.8%) | 24 (6.1%) | 31 (10.0%) | 0.15 |
| **BMI** | | | | | | | |
| Students who were underweight (<-2SD from median for BMI by age and sex) | 98 (12.9%) | 55 (13.7%) | 43 (12.1%) | 68 (9.7%) | 47 (12.0%) | 21 (6.7%) | 0.24 |
| Students who were overweight (>+1SD from median for BMI by age and sex) | 103 (13.6%) | 58 (14.5%) | 45 (12.6%) | 107 (15.2%) | 65 (16.6%) | 42 (13.5%) | 0.25 |
| Students who were obese (>+2SD from median for BMI by age and sex) | 40 (5.3%) | 25 (6.2%) | 15 (4.2%) | 53 (7.5%) | 32 (8.1%) | 21 (6.7%) | 0.96 |
| **Hygiene** | | | | | | | |
| Students who brushed their teeth 2 times per day | 417 (55.3%) | 211 (52.8%) | 206 (58.0%) | 383 (54.7%) | 211 (53.9%) | 172 (55.6%) | 0.82 |
| Students who always or most of time washed their hands before eating | 658 (87.2%) | 352 (88.2%) | 306 (86.1%) | 585 (83.5%) | 329 (84.1%) | 256 (82.8%) | 0.04 |
| Students who always or most of time washed their hands after using the toilet/latrine | 701 (92.9%) | 372 (92.4%) | 329 (91.8%) | 630 (90.0%) | 356 (91.0%) | 274 (88.6%) | 0.04 |
| **Physical Activity** | | | | | | | |
| Students who were physically active for a total of at least 60 minutes per day on all 7 days during the past 7 days | 497 (66.1%) | 259 (65.0%) | 238 (67.0%) | 475 (67.8%) | 261 (66.9%) | 214 (69.4%) | 0.44 |
| Students who spent three or more hours per day sitting and watching or doing other sitting activities | 109 (14.4%) | 57 (14.3%) | 52 (14.6%) | 117 (16.7%) | 74 (18.9%) | 43 (13.9%) | 0.24 |
| **Protective Factors** | | | | | | | |
| Students who missed classes or school without permission | 168 (22.2%) | 76 (19.0%) | 92 (25.9%) | 155 (22.1%) | 87 (22.2%) | 68 (22.0%) | 0.88 |
| Students who reported that most of the students in their school were never or rarely kind and helpful | 280 (37.2%) | 148 (37.1%) | 132 (37.2%) | 310 (44.4%) | 166 (42.4%) | 144 (46.9%) | 0.01 |
| students whose parents or guardians never or rarely really knew what they were doing with their free time | 287 (38.0%) | 145 (36.3%) | 142 (40%) | 303 (43.2%) | 161 (41.1%) | 142 (45.9%) | 0.04 |

schools for health profile assessment outcomes. The authors found an association between higher accreditation and intervention time period of year for both government and private schools (Table 2). The study showed an improvement in number of government and private schools in two of eight accreditation domains associated with the intervention (Table 3), higher hygiene levels and protective factors with higher accredited schools (Table 4), and mental health improvements but poorer sexual health awareness associated with higher accredited schools (Table 5).

School settings have long been advocated as an excellent health promoting settings. Studies have reported that evidence-based interventions in the school setting should be promoted as an important component for integrated programs, policies, and monitoring frameworks designed to reverse the childhood obesity in the region [8]. Global standards for HPS [9] has been launched recently which has also emphasized effectiveness of HPS. Present study is an effort to assess effectiveness of HPS in Indian settings. The accreditation scheme has been implemented only in Thailand in the Southeast Asia region. In India, Chandigarh has become the first city to have undergone accreditation of all schools as HPS by voluntary and

**Table 5. Comparison of key behaviours of (tobacco, alcohol, drug abuse, violence) and mental health among school children in higher and lower accreditation schools in Chandigarh.**

| Results for students aged 11–16 years | Higher Accredited Schools | | | Lower Accredited Schools | | | Pvalue |
|---|---|---|---|---|---|---|---|
| | Total N(754) n% | Boys N(399) n% | Girls N(355) n% | Total N(700) n% | Boys N(391) n% | Girls N(309) n% | |
| **Tobacco Use** | | | | | | | |
| Students who smoked cigarettes on one or more days | 16 (2.1%) | 7 (1.7%) | 9 (2.5%) | 16 (2.2%) | 5 (1.2%) | 11 (3.5%) | 0.83 |
| Students who used any tobacco products other than cigarettes on one or more days | 18 (2.3%) | 9 (2.2%) | 9 (2.5%) | 20 (2.8%) | 9 (2.3%) | 11 (3.5%) | 0.57 |
| Students who reported people smoking in their presence on one or more days during the past 7 days | 230 (30.5%) | 123 (30.9%) | 107 (30.1%) | 209 (29.8%) | 121 (30.9%) | 88 (28.4%) | 0.8 |
| **Mental health** | | | | | | | |
| Students who have been so worried about something that they wanted to use alcohol never or rarely alcohol or other drugs to feel better | 724 (96.0%) | 383 (95.9%) | 341 (96.0%) | 668 (95.4%) | 370 (95.1%) | 298 (96.7%) | 0.76 |
| Students who were taught in any of their classes how to handle stress in healthy ways during this school year | 472 (62.5%) | 247 (62.0%) | 225 (63.3%) | 386 (55.1%) | 235 (60.2%) | 151 (49.1%) | 0.01 |
| **Violence And Unintentional Injury** | | | | | | | |
| students who were taught how to avoid being bullied in their classes | 399 (52.9%) | 399 (100%) | 0(0%) | 391 (55.8%) | 391 (100%) | 0 (0%) | 0.26 |
| students who reported that they had serious injury happened to them | 466 (61.8%) | 227 (56.8%) | 239 (67.3%) | 463 (66.1%) | 237 (60.6%) | 226 (73.1%) | 0.07 |
| **Alcohol Use Module** | | | | | | | |
| students who have been taught dangers of alcohol use in school year | 327 (43.3%) | 174 (43.6%) | 153 (43.0%) | 326 (46.5%) | 188 (48.0%) | 138 (44.6%) | 0.19 |
| students who reported that parents or guardians drink alcohol | 153 (20.2%) | 75 (18.7%) | 78 (21.9%) | 114 (16.2%) | 70 (17.9%) | 44 (14.3%) | 0.05 |
| **Drugs Use Module** | | | | | | | |
| students during the past 12 months, how much times have they used drugs | 23 (3.0%) | 10 (2.5%) | 13 (3.6%) | 25 (3.5%) | 17 (4.3%) | 8 (2.5%) | 0.57 |
| students during their school year, who taught in any of their classes the problems associated with using drugs | 281 (37.2%) | 145 (36.3%) | 136 (38.3%) | 255 (36.4%) | 138 (35.2%) | 117 (37.8%) | 0.756 |
| **Sexual Behavior Module** | | | | | | | |
| Students who were taught in any of their classes about HIV infection or AIDS in school year | 271 (35.9%) | 126 (31.5%) | 145 (40.8%) | 320 (45.7%) | 168 (42.9%) | 152 (49.1%) | 0.01 |
| students who heard of HIV infections or AIDS | 394 (52.2%) | 199 (49.8%) | 195 (54.9%) | 383 (54.7%) | 201 (51.4%) | 182 (58.8%) | 0.43 |
| Students who were taught in any of their classes how to avoid HIV infection or AIDS in school year | 251 (33.2%) | 123 (30.8%) | 128 (36.0%) | 251 (35.8%) | 134 (34.2%) | 117 (37.8%) | 0.29 |
| Students who talked about HIV infection or AIDS with their parents or guardians | 168 (22.2%) | 70 (17.5%) | 98 (27.6%) | 159 (22.7%) | 72 (18.4%) | 87 (28.1%) | 0.83 |
| students who reported that neatly looking person can be infected with HIV | 174 (23.0%) | 86 (21.5%) | 88 (24.7%) | 213 (30.4%) | 120 (30.6%) | 93 (30.0%) | 0.01 |

continuous quality improvement process under different parameters. The competitive spirit was amply observed in the present study on achieving particular accreditation level and ambition to move to the next level was quite evident. There was significant improvement in accreditation level of schools as HPS by self-quality improvement with technical support from key stakeholders over a period of 1 year of the intervention.

Study conducted in Hong Kong found that students in HPSs had a more positive health behavior profile as compared to those in non-HPS [10]. A review evaluating nine studies of HPS emphasized that HPS has some influence on various domains of health for the school community [11]. Another study highlighted there is scope for integrating health promotion into school policies and the curriculum [12]. A multi-component model of nutrition and life-style education was found to be effective in improving the nutrition-related knowledge, life-style practices, and resulted in beneficial changes in anthropometric and biochemical profiles of the Asian Indian adolescents [13].

In the current study, it was observed that knowledge and practices of personal hygiene of students studying in higher accredited schools were better than students of lower accredited schools. A study reported that low awareness about personal hygiene were the key areas of concern and could be tackled by the active involvement of school teachers, bringing about improvement in personal hygiene of school children [14]. Personal hygiene, mental health have been identified as priority areas for HPS. Hence students of higher accredited schools in the present study were more attentive to their personal hygiene and were better equipped to handle stress in schools.

Our study demonstrated that Shaala Siddhi has been implemented in government schools as compared to private schools. Shaala Siddhi is a national programme on school standards and evaluation [15]. It is a school self-evaluation process in a sequential manner and particu-larly domains 1,5,6,7 i.e., enabling resources of the school, school leadership and management, inclusion, health and safety, and productive community participation are related to HPS.

Our study showed that community partnership in decision-making and planning in the health-promoting activities improved after one year of intervention in the current study. The HPS concept highlights community participation as integral to the success of health-promot-ing interventions. Studies have reported that ownership, leadership, and collaboration are crit-ical to improving school health [14].

Students from lower accredited schools were more informed about sexual behaviour mod-ule. Most of the lower accredited schools were in the periphery area of the Chandigarh city, considered at higher risk for the prevalence of HIV which are covered under targeted interven-tions projects for HIV (TI project). Hence the extent of interventions on HIV might be more in these schools.

Schools are designated as settings that can help reduce inequalities in health through net-working with other stakeholders [5]. Literature has suggested that HPS activities primarily focus on student health [16] and improvement in the health promoting behaviour of the stu-dents. It has also been reported that accreditation define dedication and acknowledge excel-lence; this may encourage schools to improve and become HPS which has been observed in the present study as well [16]. The current study found that HPS provided ground for better coordination between health and education department and other sectors.

## Limitation of the study

Health profile assessment could not be performed for all included schools due to time and budget constraint. Comparison of accredited and non-accredited schools was not an objective of the project hence not presented in the current paper.

## Conclusions

The accreditation system of schools as HPS was found to be feasible and effective as there was significant improvement in accreditation level after continuous self-quality improvement by schools and better health profile of children studying in higher accredited schools as compared

to low accredited schools and should be promoted. assessment and classification of schools on accreditation levels in school settings is feasible and must be integrated into the school education system. Considering the results of the study assessment for accreditation of schools as HPS should be conducted on 3–5 years basis for comparative report and further improvement.

## Supporting information

**S1 File.**
(XLS)

## Acknowledgments

We acknowledge the contributions of HPS project staff who provided general support.

## Author Contributions

**Conceptualization:** Jarnail Singh Thakur, Bhavneet Bharti, Meenakshi Sharma.

**Data curation:** Meenakshi Sharma, Rupinder Kaur.

**Formal analysis:** Meenakshi Sharma, Rupinder Kaur.

**Funding acquisition:** Jarnail Singh Thakur.

**Methodology:** Jarnail Singh Thakur, Sukriti Singh, Bhavneet Bharti, Rupinder Kaur.

**Project administration:** Jarnail Singh Thakur, Meenakshi Sharma.

**Resources:** Jarnail Singh Thakur.

**Supervision:** Jarnail Singh Thakur, Meenakshi Sharma, Meenakshi Sharma.

**Validation:** Jarnail Singh Thakur, Meenakshi Sharma.

**Visualization:** Jarnail Singh Thakur, Sukriti Singh, Meenakshi Sharma.

**Writing – original draft:** Jarnail Singh Thakur, Meenakshi Sharma, Rupinder Kaur.

**Writing – review & editing:** Jarnail Singh Thakur, Sukriti Singh, Bhavneet Bharti, Meenakshi Sharma.

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
