## [Decision Letter · Decision Letter 0]

10 Aug 2021

PONE-D-21-05043

Effectiveness of Health Promoting Schools: A Comparative Health Profile Assessment of higher as compared to low accredited Schools in Chandigarh,Union Territory of North India

PLOS ONE

Dear Dr. Thakur,

Thank you for submitting your manuscript to PLOS ONE. After careful consideration, we feel that it has merit but does not fully meet PLOS ONE’s publication criteria as it currently stands. Therefore, we invite you to submit a revised version of the manuscript that addresses the points raised during the review process.

The reviewers have identified a number of concerns regarding the presentation of your work, and have requested clarification of several important aspects of your methods. Please attend carefully to each of the points they have raised when preparing your revisions.

We look forward to receiving your revised manuscript.

Kind regards,

Jamie Males

Staff Editor

PLOS ONE

Journal Requirements:

A clean copy of the edited manuscript (uploaded as the new *manuscript* file)”.

3. Thank you for including your ethics statement:  "Ethical permission for conduction of study was taken from the institutional ethics committee."

Reviewers' comments:

Reviewer's Responses to Questions

**Comments to the Author**

1. Is the manuscript technically sound, and do the data support the conclusions?

Reviewer #1: Partly

Reviewer #2: Yes

2. Has the statistical analysis been performed appropriately and rigorously? 

Reviewer #1: Yes

Reviewer #2: Yes

3. Have the authors made all data underlying the findings in their manuscript fully available?

Reviewer #1: No

Reviewer #2: Yes

4. Is the manuscript presented in an intelligible fashion and written in standard English?

Reviewer #1: Yes

Reviewer #2: No

5. Review Comments to the Author

Reviewer #1: Study on Effectiveness of Health Promoting School (HPS) is very much needed. I commend the authors for the dedicated effort to gain new knowledge to evidence of HPS effectiveness. However, more work is needed for publication.

There should be more description of the accreditation of different levels of awards and the process of validation. Comparison of basic characteristics of schools with different levels of award and non-award schools should be performed to analyse any difference at baseline. Those would be confounding factors.

The authors need to jsutify the categorisation between higher accreditation schools (Gold and Silver- 1+9=10) amd lower accreditionschools (Bronze or below bronze-24+152=176). The standard below bronze would be variable and the scientific basis of putting bronze and below bronze in same category needs to be justified.

Health profile of students from 8 higher accredition schools with 754 students were compared with 700 students from lower acrreditation ? numbers of schools. There is no description of sampling methods and the rationale behind. There were 152 lower accreditiation schools and this reference group can have larger sample size. If all schools with award are categorised as accedition schools, there will be students from 34 schools comparing with students from 152 non-accredition schools, this would allow better analysis of effect size. The Hong Kong studies published ib 2006 and 2008 cited by the authors categories schools into award and non-award groups.

Apart from oberall analysisi of pre and post interventions, authors can conduct separate pre and post interventions among accreditation schools and non-accreditiation schools. For comparison of health profiles, those factors. showing statistical signifiance, most of them only demonstrated less than 10% difference and many only around 5%. Authors would consider re-analysis with bronze schools to be included under acceditiation schools comparing with non-accreditation schools. If data available, should include all students from those schools and it will give rise to much larger sample size.

Multiple logisitic regression can be considered to analyse the factor(s) including accreditation status together with other independent variables predicting particular health outcomes. This will allow controlling the confounding factors. Those components under different domains showing statistical difference from pre and post analysis would also be included as independent variables to analyse which component(s) is independent predictor for better health profile.

The authors should draw on more recent literature on HPS to strenghten the discussuon section. The speical issue of Health Promotion International on HPS in 2017 and other recent papers.

Reviewer #2: General Comments

This is a pre-post multicomponent and multilevel health promotion school accreditation intervention study of both government and private secondary schools in Northern India. The aim was to compare the higher accredited schools versus lower accredited schools for health profile assessment outcomes. The authors showed an association between higher accreditation levels and time period (table 2), increase in two of eight accreditation domains associated with the intervention (table 3), higher hygiene levels and protective factors with higher accredited schools (table 4), and mental health improvements but poorer sexual health awareness associated with higher accredited schools (table 5). To improve the manuscript readability, the paper needs to be edited for conciseness and follow a more structured approach. It is not necessary to quote percentage to 2 decimal places in Tables-1 decimal place is sufficient. Be consistent with number of decimal places for P values throughout manuscript.

Specific Comments

1. Title needs to be more informative. Possible suggestion “Association between health promotion school accreditation levels and health profile assessment over time in Northern India.” The use of effectiveness implies randomization.

2. Ethical statement. This need to be completed. Please specify name of ethics committee, reference number and date of approval. Also include these details in line 156.

3. Introduction. Lines 83 to 105. This is a long paragraph. Suggest paragraph break at line 93 to improve readability.

4. Consider a subheading ‘Intervention’ so it is clear what it is before line 116.

5. Line 121. Bolded ‘Manual’. Is this subheading or part of a sentence?

6. Line 128. Bolded ‘Key school-based intervention’. Is this subheading or part of a sentence?

7. Line 131. Social media does not need to be bolded.

8. Line 133. Bolded ‘Orientation and reorientation’ subheading or sentence?

9. Line 141. Bolded ‘The health profile assessment’ subheading or sentence?

10. Line 145-147. Specify exactly how schools were randomly selected.

11. Write out in full abbreviations ‘CBSE’ (line 149), indicate what SHAALA SIDDHI in Table 3 means with footnote.

12. Line 149 Include reference

13. Line 161. Specify version, company and city for SPSS and Epiinfo.

14. State significant level P<0.05 and 2-sided significance in data analysis.

15. Text in Results is too long. Much of it is repeating the information in the Tables. Consider adding a flow diagram of schools, like STROBE, to help reader understand the study design and data analysis.

16. Table 1. Median (IQR) for sex characteristics are not clear. Is this number of girls and boys in each school?

17. Table 2. Column for preintervention private school column does not add up to 100%. Consider conducting a Cochrane-Armitage test for trend for total pre and post intervention accreditation level, which would be significant to show that overall accreditation levels changed over time. %age change should be rewritten to Change with a footnote to indicate percentage; consider 95% CI around percentage change.

18. Table 3. Title Score �3 is unclear to readers. Please explain.

19. Table 4. Check all P value, especially for BMI first row and third row, should this be marginally significant?

20. The discussion was not concise and structured. First paragraph should highlight main results as indicated in general comments above. Then following paragraphs should discussed each of the main results in sequence with Table order in comparison with other published studies. There should be a study limitation paragraph.

6. PLOS authors have the option to publish the peer review history of their article (what does this mean?). If published, this will include your full peer review and any attached files.

Reviewer #1: **Yes: **Professor Albert Lee

Reviewer #2: No

---

## [Author Response · Author response to Decision Letter 0]

23 Jan 2022

1. Plos file naming forma have been ensued

2. Copyright and editing done by senior authors of the manuscript

3.Ethics statement added with the ethics commitee approval number

4.The study has been conducted under the project funded by Indian Council of Medical Research. Funding was provided for conducting the study, salaries of the project staff, travel and contingency grant. But no funding assistance was provided for publications of the manuscript.

5. All data underlying the findings in the manuscript fully available on request to Principal investigator

---

## [Decision Letter · Decision Letter 1]

3 May 2022

PONE-D-21-05043R1

Effectiveness of Health Promoting Schools: A Comparative Health Profile Assessment of higher as compared to low accredited Schools in Chandigarh,Union Territory of North India

PLOS ONE

Dear Dr. Thakur,

Thank you for submitting your manuscript to PLOS ONE. After careful consideration, we feel that it has merit but does not fully meet PLOS ONE’s publication criteria as it currently stands. Therefore, we invite you to submit a revised version of the manuscript that addresses the points raised during the review process.

We look forward to receiving your revised manuscript.

Kind regards,

Amitava Mukherjee, ME, Ph.D.

Academic Editor

PLOS ONE

Journal Requirements:

Reviewers' comments:

Reviewer's Responses to Questions

**Comments to the Author**

1. If the authors have adequately addressed your comments raised in a previous round of review and you feel that this manuscript is now acceptable for publication, you may indicate that here to bypass the “Comments to the Author” section, enter your conflict of interest statement in the “Confidential to Editor” section, and submit your "Accept" recommendation.

Reviewer #2: All comments have been addressed

2. Is the manuscript technically sound, and do the data support the conclusions?

Reviewer #2: Yes

3. Has the statistical analysis been performed appropriately and rigorously? 

Reviewer #2: Yes

4. Have the authors made all data underlying the findings in their manuscript fully available?

Reviewer #2: No

5. Is the manuscript presented in an intelligible fashion and written in standard English?

Reviewer #2: Yes

6. Review Comments to the Author

Reviewer #2: The revised manuscript reads much better. Just further minor amendments to be made:

1. Move limitation of the study to before conclusion section

2. Provide exact grant number of the government funding agency grant you obtained for completeness

3. Relabel 'Figure 2 Strobes flow diagram' to 'Figure 2 Participating Schools Flow Diagram'

7. PLOS authors have the option to publish the peer review history of their article (what does this mean?). If published, this will include your full peer review and any attached files.

Reviewer #2: No

---

## [Author Response · Author response to Decision Letter 1]

17 Jun 2022

We are thankful for your valuable comments. All comments suggested have been addressed

---

## [Editor Report · Decision Letter 2]

20 Jun 2022

Effectiveness of Health Promoting Schools: A Comparative Health Profile Assessment of higher as compared to low accredited Schools in Chandigarh,Union Territory of North India

PONE-D-21-05043R2

Dear Dr. Thakur,

We’re pleased to inform you that your manuscript has been judged scientifically suitable for publication and will be formally accepted for publication once it meets all outstanding technical requirements.

Kind regards,

Amitava Mukherjee, ME, Ph.D.

Academic Editor

PLOS ONE
---

## [Editor Report · Acceptance letter]

21 Sep 2022

PONE-D-21-05043R2 

Effectiveness of Health Promoting Schools: A Comparative Health Profile Assessment of higher as compared to low accredited Schools in Chandigarh, Union Territory of North India 

Dear Dr. Thakur:

I'm pleased to inform you that your manuscript has been deemed suitable for publication in PLOS ONE. Congratulations! Your manuscript is now with our production department. 

Kind regards, 

on behalf of

Professor Dr. Amitava Mukherjee 

Academic Editor

PLOS ONE